# Stability Results for a Weakly Dissipative Viscoelastic Equation with Variable-Exponent Nonlinearity: Theory and Numerics

**Adel M. Al-Mahdi** [1,2,*], **Mohammad M. Al-Gharabli** [1,2], **Maher Noor** [3] **and Johnson D. Audu** [1]

1   The Preparatory Year Program, King Fahd University of Petroleum and Minerals, Dhahran 31261, Saudi Arabia
2   The Interdisciplinary Research Center in Construction and Building Materials, King Fahd University of Petroleum & Minerals, Dhahran 31261, Saudi Arabia
3   DCC-Math, King Fahd University of Petroleum and Minerals, Dhahran 31261, Saudi Arabia
*   Correspondence: almahdi@kfupm.edu.sa

**Abstract:** In this paper, we study the long-time behavior of a weakly dissipative viscoelastic equation with variable exponent nonlinearity of the form $u_{tt} + \Delta^2 u - \int_0^t g(t-s)\Delta u(s)ds + a|u_t|^{n(\cdot)-2}u_t - \Delta u_t = 0$, where $n(.)$ is a continuous function satisfying some assumptions and $g$ is a general relaxation function such that $g'(t) \leq -\xi(t)\mathbb{G}(g(t))$, where $\xi$ and $\mathbb{G}$ are functions satisfying some specific properties that will be mentioned in the paper. Depending on the nature of the decay rate of $g$ and the variable exponent $n(.)$, we establish explicit and general decay results of the energy functional. We give some numerical illustrations to support our theoretical results. Our results improve some earlier works in the literature.

**Keywords:** viscoelasticity; relaxation function; general decay; convex functions; variable exponent; numerical computations





## 1. Introduction

In this paper, we consider the following weakly dissipative viscoelastic equation with variable exponent nonlinearity

$$u_{tt} + \Delta^2 u - \int_0^t g(t-s)\Delta u(s)ds + a|u_t|^{n(\cdot)-2}u_t - \Delta u_t = 0 \text{ in } \Omega \times (0,+\infty), \quad (1)$$

subject to the following conditions

$$\begin{cases} u = \Delta u = 0, & \text{on } \partial\Omega, \\ u(x,0) = u_0(x), \ u_t(x,0) = u_1(x), & \text{in } \Omega, \end{cases} \quad (2)$$

where $\Omega$ is a bounded domain of $\mathbb{R}^d$ with a smooth boundary $\partial\Omega$, $a$ is a positive constant, $n(.)$ is a continuous function satisfying some assumptions, $g$ is a general relaxation function satisfying some conditions and $(u_0, u_1)$ are the given initial data.

For the stabilization of weakly dissipative second-order systems, Rivera et al. [1] considered the following abstract integro-differential equation

$$u_{tt} + \mathcal{A}u + \beta u - \int_0^t g(t-s)\mathcal{A}^\alpha u ds = 0, \quad (3)$$

where $\mathcal{A}$ is a strictly positive self-adjoint linear operator and established a polynomial decay result with the interpolating cases $\alpha \in (0,1)$ and $g$ decays exponentially to zero.

In [2], Hassan and Messaoudi generalized the result of [1] by considering a very general assumption on the relaxation function $g$ and obtained a new general decay rate. In [3], Anaya and Messaoudi discussed the following problem

$$u_{tt} + \Delta^2 u - \int_0^t g(t-s)\Delta u(s)ds + h(u_t) = 0, \tag{4}$$

with a general weak damping term, and they derived general decay rate estimates under certain restrictions for the relaxation function $g$, and the function $h$ is a nondecreasing $C^1(\mathbb{R})$ function satisfying $h(0) = 0$, $sh(s) \geq 0$ and for $c_1, c_2 > 0$,

$$c_1|s| \leq |h(s)| \leq c_2|s|, \ s \in \mathbb{R}. \tag{5}$$

In recent years, there has been increasing interest in treating equations with variable exponents of nonlinearity. Some models from physical phenomena such as flows of electro-rheological fluids or fluids with temperature-dependent viscosity, filtration processes in porous media, nonlinear viscoelasticity, and image processing give rise to such problems. This great interest is motivated by the applications to the mathematical modeling of non-Newtonian fluids. One of these fluids is the electro-rheological fluids which have the ability to drastically change when applying some external electromagnetic field. The variable exponent of nonlinearity is a given function of density, temperature, saturation, electric field, etc. For more information about the mathematical model of electro-rheological fluids, we refer to [4,5]. However, there are few available works in the literature including nonlinearities of variable-exponent type. For example, Antontsev [6,7] considered the following problem

$$u_{tt} - div(a(x)|\nabla u|^{p(x,t)-2}\nabla u) - \alpha\Delta u_t = b(x,t)|u|^{\sigma(x,t)-2}u, \tag{6}$$

where $\alpha$ is a positive constant and the author established local, global existence and blow-up results under some conditions on the functions $a, b, p, \sigma$. In [8], Problem (6) was also considered, and the authors proved various blow-up results for the solutions with positive initial energy. Furthermore, in [9], the authors studied the following equation

$$u_{tt} - div(|\nabla u|^{r(x)-2}\nabla u) + a|u_t|^{n(x)-1}u_t = b|u|^{p(x)-2}u, \tag{7}$$

and proved the existence of a unique weak solution. The authors of [9] also established a finite-time blow up of the solutions. In [10], Messaoudi et al. considered Problem (7), where $b = 0$, and established decay estimates for the solutions under suitable assumptions on the initial data and the variable exponents.

Recently, in [11], Messaoudi investigated the following problem

$$u_{tt} - div(a(x)|\nabla u|^{r(.)-2}\nabla u) - \Delta u_t + |u_t|^{n(.)-2}u_t = 0, \tag{8}$$

and established exponential and polynomial decay results under some conditions on the variable exponents $n$ and $r$. The existence, stability and blow up of solutions of other problems with variable exponents such as Petrovsky, Kirchhoff and other viscoelastic equations can be found in the references [12–16].

In the present work, we are interested in establishing explicit and general decay results for the system (1)–(2) by using the energy method and some properties related to the variable exponents. Then, we give some numerical examples to illustrate our results. Our decay results depend on the nature of the decay rate of the relaxation function $g$ and the variable exponent $n(.)$. The results improve the recent works in [1–3], where the authors considered the usual constant exponents.

The remainder of this paper is organized as follows: In Section 2, we outline some preliminaries. In Section 3, we state and prove some essential lemmas which are needed in the proofs of the decay results. In Section 4, we establish the general decay results of

our problem (1)–(2) and present some examples. Finally, in Section 5, we provide some numerical examples to illustrate our theoretical results.

## 2. Preliminaries

In this work, $L^2(\Omega)$ denotes the standard Lesbesgue space, and we define following

$$H(\Omega) := \{u \in H^3(\Omega) : u = \Delta u = 0 \text{ on } \partial\Omega\}$$

is a Sobolev space with the usual scalar products and norms. Throughout this paper, $k$ is a generic positive constant. Details on Lebesgue and Sobolev spaces with variable exponents can be found in [17–19]). Here, we provide some basic definitions. Let $q : \Omega \to [1, \infty]$ be a measurable function, where $\Omega$ is a domain of $\mathbb{R}^d$. The Lebesgue space with a variable exponent $q(\cdot)$ is given by

$$L^{q(\cdot)}(\Omega) := \left\{ u : \Omega \to \mathbb{R}; \text{ measurable in } \Omega : \rho_{q(\cdot)}(\mu u) < \infty, \text{ for some } \mu > 0 \right\},$$

where

$$\rho_{q(\cdot)}(u) = \int_\Omega |u(x)|^{q(x)} dx.$$

The space $L^{q(\cdot)}(\Omega)$ is Banach (see [18]) when equipped with the norm

$$\|u\|_{q(\cdot)} := \inf \left\{ \mu > 0 : \int_\Omega \left| \frac{u(x)}{\mu} \right|^{q(x)} dx < \infty \right\}, \quad \forall u \in L^{q(\cdot)}(\Omega).$$

Furthermore, $L^{q(\cdot)}(\Omega)$ is both reflexive and separable for $q_2 \in (q_1, \infty)$, provided $q(\cdot)$ is bounded with

$$q_1 := \text{essinf}_{x \in \Omega} q(x), \quad q_2 := \text{esssup}_{x \in \Omega} q(x).$$

The variable-exponent Sobolev space defined by

$$W^{1,q(\cdot)}(\Omega) = \left\{ u \in L^{q(\cdot)}(\Omega) \text{ such that } \nabla u \text{ exists and } |\nabla u| \in L^{q(\cdot)}(\Omega) \right\}$$

is also Banach when equipped with the norm $\|u\|_{W^{1,q(\cdot)}(\Omega)} = \|u\|_{q(\cdot)} + \|\nabla u\|_{q(\cdot)}$. What is more is that $W^{1,q(\cdot)}(\Omega)$ is separable and reflexive for $1 < q_1 \leq q_2 < \infty$, provided $q(\cdot)$ is bounded.

**Lemma 1** ([18]). *Let $\Omega$ be a bounded domain in $\mathbb{R}^d$ with a smooth boundary $\partial\Omega$. Assume that $p, q \in C(\overline{\Omega})$ such that for all $x \in \overline{\Omega}$,*

$$1 < p_1 \leq p(x) \leq p_2 < +\infty, \qquad 1 < q_1 \leq q(x) \leq q_2 < +\infty, \quad ,$$

*and $q(x) < p^*(x) \in \overline{\Omega}$ with $p^*(x) =$*

$$\begin{cases} \frac{dp^*(x)}{d - p^*(x)}, & if \quad p_2 < d, \\ +\infty, & if \quad p_2 \geq d, \end{cases}$$

*then $W^{1,p(\cdot)}(\Omega)$ is continuously and compactly embedded in $L^{q(\cdot)}(\Omega)$. Consequently, there exists a constant $k_e > 0$ such that*

$$\|u\|_q \leq k_e \|u\|_{W^{1,p(\cdot)}}, \quad \forall u \in W^{1,p(\cdot)}(\Omega).$$

In this work, we assume the following:

(C1) The function $g : \mathbb{R}^+ \to \mathbb{R}^+$ is a $C^1$ non-increasing function satisfying

$$g(0) > 0, \quad 1 - \omega_0 \int_0^\infty g(s)ds = \ell > 0 \tag{9}$$

where

$$\|\nabla u\|_2^2 \leq \omega_0 \|\Delta u\|_2^2, \ \forall u \in H(\Omega), \tag{10}$$

and there exists a $C^1$ function $\mathbb{G} : (0, \infty) \to (0, \infty)$ which is strictly increasing and a strictly convex $C^2$ function on $(0, r], r \leq g(0)$, with $\mathbb{G}(0) = \mathbb{G}'(0) = 0$, such that

$$g'(t) \leq -\xi(t)\mathbb{G}(g(t)), \quad \forall t \geq 0, \tag{11}$$

where $\xi$ is a positive non-increasing differentiable function.

(C2) $n : \overline{\Omega} \to [1, \infty)$ is a continuous function such that

$$n_1 := \operatorname{essinf}_{x \in \Omega} n(x), \quad n_2 := \operatorname{esssup}_{x \in \Omega} n(x).$$

and $1 < n_1 \leq n(x) \leq n_2$, where

$$\begin{cases} n_2 < \infty, & d = 1, 2; \\ n_2 \leq \frac{2d}{d-2}, & d \geq 3. \end{cases}$$

Furthermore, the exponent $n(\cdot)$ satisfies the log-Hölder continuity condition; that is

$$|n(x) - n(y)| \leq -\frac{A}{\log|x-y|}, \text{ for all } x, y \in \Omega, \text{ with } |x - y| < \delta, \tag{12}$$

for any $A, \delta > 0$.

The energy functional associated to (1)–(2) is given by

$$\begin{aligned} \mathcal{E}(t) &= \frac{1}{2}\left[\|u_t\|_2^2 + \|\Delta u\|_2^2 - \left(\int_0^t g(s)ds\right)\|\nabla u\|_2^2 + (g \circ \nabla u)(t)\right] \\ &\geq \frac{1}{2}\left[\|u_t\|_2^2 + \left(1 - \omega_0\int_0^t g(s)ds\right)\|\Delta u\|_2^2 + (g \circ \nabla u)(t)\right] \geq 0, \end{aligned} \tag{13}$$

for any $t \geq 0$, where for $v \in L^2_{loc}(\mathbb{R}^+; L^2(\Omega))$,

$$(g \circ v)(t) := \int_0^t g(t-s)\|v(t) - v(s)\|_2^2 ds.$$

**Lemma 2.** *For any $t \geq 0$, the energy functional $\mathcal{E}(t)$ satisfies*

$$\mathcal{E}'(t) = \frac{1}{2}(g' \circ \nabla u)(t) - \frac{1}{2}g(t)\|\nabla u\|_2^2 - a\int_\Omega |u_t|^{n(x)}dx - \|\nabla u_t\|_2^2 \leq 0. \tag{14}$$

**Proof.** In view of the boundary conditions (2), we can deduce (14) simply by multiplying (1) by $u_t$ and integrate over $\Omega$. □

**Remark 1** ([20]). *Using (C1), one can show that for any $t \in [0, t_0]$ and for some $\kappa > 0$,*

$$g'(t) \leq -\xi(t)\mathbb{G}(g(t)) \leq -\kappa\xi(t) = -\frac{\kappa}{g(0)}\xi(t)g(0) \leq -\frac{\kappa}{g(0)}\xi(t)g(t)$$

*and hence,*

$$\xi(t)g(t) \leq -\frac{g(0)}{\kappa}(t), \quad \forall t \in [0, t_0]. \tag{15}$$

*Moreover, if $\mathbb{G}$ is a strictly increasing and strictly convex $C^2$ function on $(0, r]$, with $\mathbb{G}(0) = \mathbb{G}'(0) = 0$, then there is a strictly convex and strictly increasing $C^2$ function $\bar{\mathbb{G}} : [0, +\infty) \longrightarrow [0, +\infty)$ which is an extension of $\mathbb{G}$. For instance, we can define $\bar{\mathbb{G}}$, for any $t > r$, by*

$$\bar{\mathbb{G}}(t) := \frac{\mathbb{G}''(r)}{2}t^2 + (\mathbb{G}'(r) - \mathbb{G}''(r)r)t + \left(\mathbb{G}(r) + \frac{\mathbb{G}''(r)}{2}r^2 - \mathbb{G}'(r)r\right).$$

## 3. Technical Lemmas

We establish some important lemmas in this section.

**Lemma 3** ([21])**.** *Under Assumption* (C1)*, we have for any $v \in L^2_{loc}\big([0, +\infty); L^2(0, L)\big)$,*

$$\int_0^L \left(\int_0^t g(t-s)(v(t) - v(s))ds\right)^2 dx \leq K_\varepsilon (h_\varepsilon \circ v)(t), \qquad \forall\, t \geq 0. \tag{16}$$

*where*

$$K_\varepsilon := \int_0^\infty \frac{g^2(s)}{\varepsilon g(s) - g'(s)}ds \qquad \text{and} \qquad h_\varepsilon(t) := \varepsilon g(t) - g'(t),$$

*for any $0 < \varepsilon < 1$.*

**Lemma 4.** *Under the assumptions* (C1) *and* (C2)*, and for $0 < \delta < 1$, the functional*

$$\mathbb{I}(t) := \int_\Omega u u_t dx$$

*satisfies the estimates:*

$$\mathbb{I}'(t) \leq \|u_t\|_2^2 + \frac{2\omega_0}{\ell}\|\nabla u_t\|_2^2 - \frac{\ell}{4}\|\Delta u\|_2^2 + k\, K_\varepsilon(h_\varepsilon \circ \nabla u)(t) \\ + \int_\Omega k_\delta(x)|u_t|^{n(x)}dx, \quad n_1 \geq 2. \tag{17}$$

*and if $1 < n_1 < 2$,*

$$\mathbb{I}'(t) \leq \|u_t\|_2^2 + \frac{3\omega_0}{\ell}\|\nabla u_t\|_2^2 - \frac{\ell}{4}\|\Delta u\|_2^2 + k\, K_\varepsilon(h_\varepsilon \circ \nabla u)(t) \\ + k\left[\int_\Omega (1 + k_\delta(x))|u_t|^{n(x)}dx + \left(\int_\Omega |u_t|^{n(x)}\right)^{n_1-1}\right], \tag{18}$$

*where $h_\varepsilon$ is given in Lemma 3 and $k_\delta(x)$ will be given in the proof below.*

**Proof.** Differentiating $\mathbb{I}$ and using (1)–(2), we obtain

$$\mathbb{I}'(t) = \int_\Omega u_t^2 dx - \int_\Omega \nabla u.\nabla u_t dx - \|\Delta u\|_2^2 - \int_\Omega u(t)\int_0^t g(t-s)\Delta u(s)ds\, dx - a\int_\Omega u|u_t|^{n(x)-2}u_t dx. \tag{19}$$

Using Young's inequality, we have

$$-\int_\Omega \nabla u.\nabla u_t dx \leq \delta_0 \omega_0 \|\Delta u\|_2^2 + \frac{1}{4\delta_0}\|\nabla u_t\|_2^2.$$

Now, recalling (10), applying Young's inequality, using Lemma 3 and (10), we obtain

$$- \int_\Omega u(t) \int_0^t g(t-s)\Delta u(s)ds\, dx$$

$$= - \int_\Omega u(t) \int_0^t g(t-s)\big(\Delta u(s)-\Delta u(t)\big)ds\, dx - \int_\Omega u(t) \int_0^t g(t-s)\Delta u(t)ds\, dx$$

$$= \int_\Omega \nabla u(t). \int_0^t g(t-s)\big(\nabla u(s)-\nabla u(t)\big)ds\, dx + \Big( \int_0^t g(s)ds \Big) \|\nabla u\|_2^2.$$

$$\leq \Big( \int_0^t g(s)ds \Big) \|\nabla u\|_2^2 + \frac{\ell}{2\omega_0} \int_\Omega |\nabla u(t)|^2 + \frac{\omega_0}{2\ell} \int_\Omega \Big( \int_0^t g(t-s)\big(\nabla u(s)-\nabla u(t)\big)ds \Big)^2 dx. \tag{20}$$

$$\leq \omega_0 \Big( \int_0^t g(s)ds \Big) \|\Delta u\|_2^2 + \frac{\ell}{2\omega_0} \|\nabla u(t)\|_2^2 + kK_\varepsilon(h_\varepsilon \circ \nabla u)(t).$$

$$\leq (1-\ell)\|\Delta u\|_2^2 + \frac{\ell}{2}\|\Delta u(t)\|_2^2 + kK_\varepsilon(h_\varepsilon \circ \nabla u)(t).$$

$$\leq \Big(1 - \frac{\ell}{2}\Big)\|\Delta u\|_2^2 + kK_\varepsilon(h_\varepsilon \circ \nabla u)(t).$$

Applying Young's inequality with $p(x) = \frac{n(x)}{n(x)-1}$ and $p'(x) = n(x)$, we can estimate the last term in (19) as follows

$$|u_t|^{n(x)-2}u_t u \leq \delta |u|^{n(x)} + k_\delta(x)|u_t|^{n(x)}, \forall x \in \Omega,$$

where

$$k_\delta(x) = \delta^{1-n(x)}(n(x))^{-n(x)}(n(x)-1)^{n(x)-1}.$$

Hence,

$$- \int_\Omega u|u_t|^{n(x)}u_t dx \leq \delta \int_\Omega |u|^{n(x)}dx + \int_\Omega k_\delta(x)|u_t|^{n(x)}dx. \tag{21}$$

To establish (17), we set

$$\Omega_+ = \{x \in \Omega : |u(x,t)| \geq 1\} \ \text{ and } \ \Omega_- = \{x \in \Omega : |u(x,t)| < 1\}. \tag{22}$$

Then, using (10), (13), (14), (22) and Lemma 1, we obtain

$$\int_\Omega |u|^{n(x)}dx \leq \int_{\Omega_+} |u|^{n(x)}dx + \int_{\Omega_-} |u|^{n(x)}dx$$

$$\leq \int_{\Omega_+} |u|^{n_2}dx + \int_{\Omega_-} |u|^{n_1}dx \leq \int_\Omega |u|^{n_2}dx + \int_\Omega |u|^{n_1}dx$$

$$\leq \Big( k_e^{n_1}\|\nabla u\|_2^{n_1} + k_e^{n_2}\|\nabla u\|_2^{n_2} \Big)$$

$$\leq \Big( k_e^{n_1}\omega_{0\rho}^{n_1}\|\triangle u\|_2^{n_1} + k_e^{n_2}\omega_0^{n_2}\|\triangle u\|_2^{n_2} \Big) \tag{23}$$

$$\leq \Big( k_e^{n_1}\omega_0^{n_1}\|\triangle u\|_2^{n_1-2} + k_e^{n_2}\omega_0^{n_2}\|\triangle u\|_2^{n_2-2} \Big)\|\triangle u\|_2^2$$

$$\leq \Big( k_e^{n_1}\omega_0^{n_1}\Big(\frac{2\mathcal{E}(0)}{\ell}\Big)^{n_1-2} + k_e^{n_2}\omega_0^{n_2}\Big(\frac{2\mathcal{E}(0)}{\ell}\Big)^{n_2-2} \Big)\|\triangle u\|_2^2$$

$$\leq k_0\|\triangle u\|_2^2,$$

where $k_0 = \Big( k_e^{n_1}\omega_0^{n_1}\Big(\frac{2\mathcal{E}(0)}{\ell}\Big)^{n_1-2} + k_e^{n_2}\omega_0^{n_2}\Big(\frac{2\mathcal{E}(0)}{\ell}\Big)^{n_2-2} \Big).$

By fixing $\delta_0 = \frac{\ell}{8\omega_0}$ and $\delta = \frac{\ell}{8k_0}$, $k_\delta(x)$ is still bounded; hence, we obtain the required estimate (17). To prove (18), we re-write the fourth term in (19) as follows

$$- a \int_\Omega u|u_t|^{n(x)-2} u_t dx = -a \int_{\Omega_1} u|u_t|^{n(x)-2} u_t dx - a \int_{\Omega_2} u|u_t|^{n(x)-2} u_t dx, \qquad (24)$$

where

$$\Omega_1 = \{x \in \Omega : n(x) < 2\} \text{ and } \Omega_2 = \{x \in \Omega : n(x) \geq 2\}.$$

We notice that on $\Omega_1$, we have

$$2n(x) - 2 < n(x), \text{ and } 2n(x) - 2 \geq 2n_1 - 2. \qquad (25)$$

Therefore, using Young's and Poincaré's inequalities and (25), we obtain

$$
\begin{aligned}
- \int_{\Omega_1} u|u_t|^{n(x)-2} u_t dx &\leq \eta \int_{\Omega_1} |u|^2 dx + \frac{1}{4\eta} \int_{\Omega_1} |u_t|^{2n(x)-2} dx \\
&\leq \eta k_\rho^2 ||\nabla u||_2^2 + k \left[ \int_{\Omega_1^+} |u_t|^{2n(x)-2} dx + \int_{\Omega_1^-} |u_t|^{2n(x)-2} dx \right] \\
&\leq \eta k_\rho^2 \omega_0 ||\Delta u||_2^2 + k \left[ \int_{\Omega_1^+} |u_t|^{n(x)} dx + \int_{\Omega_1^-} |u_t|^{2n_1-2} dx \right] \\
&\leq \eta k_\rho^2 \omega_0 ||\Delta u||_2^2 + k \left[ \int_\Omega |u_t|^{n(x)} dx + \left( \int_{\Omega_1^-} |u_t|^2 dx \right)^{n_1-1} \right] \\
&\leq \eta k_\rho^2 \omega_0 ||\Delta u||_2^2 + k \left[ \int_\Omega |u_t|^{n(x)} dx + \left( \int_{\Omega_1^-} |u_t|^{n(x)} dx \right)^{n_1-1} \right] \\
&\leq \eta k_\rho^2 \omega_0 ||\Delta u||_2^2 + k \left[ \int_\Omega |u_t|^{n(x)} dx + \left( \int_\Omega |u_t|^{n(x)} dx \right)^{n_1-1} \right],
\end{aligned}
\qquad (26)
$$

where

$$\Omega_1^+ = \{x \in \Omega_1 : |u_t(x,t)| \geq 1\} \text{ and } \Omega_1^- = \{x \in \Omega_1 : |u_t(x,t)| < 1\}. \qquad (27)$$

Fixing $\eta = \frac{\ell}{8c_\rho^2 \omega_0}$, (26) becomes

$$- \int_{\Omega_1} u|u_t|^{n(x)-2} u_t dx \leq \frac{\ell}{8} ||\Delta u||_2^2 + c \left[ \int_\Omega |u_t|^{n(x)} dx + \left( \int_\Omega |u_t|^{n(x)} dx \right)^{n_1-1} \right]. \qquad (28)$$

Similarly, we set

$$\Omega_2^+ = \{x \in \Omega_2 : |u_t(x,t)| \in [1, \infty)\} \text{ and } \Omega_2^- = \{x \in \Omega_2 : |u_t(x,t)| \in [0, 1)\}. \qquad (29)$$

Therefore,

$$
\begin{aligned}
- a \int_{\Omega_2} u|u_t|^{n(x)-2} u_t dx &\leq \delta \int_{\Omega_2} |u|^{n(x)} dx + \int_{\Omega_2} k_\delta(x)|u_t|^{n(x)} dx \\
&\leq \delta \int_{\Omega_2^+} |u|^{n(x)} dx + \delta \int_{\Omega_2^-} |u|^{n(x)} dx + \int_{\Omega} k_\delta(x)|u_t|^{n(x)} dx \\
&\leq \delta \int_{\Omega_2^+} |u|^{n_2} dx + \delta \int_{\Omega_2^-} |u|^{n_1} dx + \int_{\Omega} k_\delta(x)|u_t|^{n(x)} dx \\
&\leq \delta \int_{\Omega} |u|^{n_2} dx + \delta \int_{\Omega} |u|^{n_1} dx + \int_{\Omega_2} k_\delta(x)|u_t|^{n(x)} dx \\
&\leq \delta \left( k_e^{n_1} ||\nabla u||_2^{n_1} + k_e^{n_2} ||\nabla u||_2^{n_2} \right) + \int_{\Omega} k_\delta(x)|u_t|^{n(x)} dx \\
&\leq \delta \left( k_e^{n_1} \omega_0^{n_1} ||\triangle u||_2^{n_1-2} + k_e^{n_2} \omega_0^{n_2} ||\triangle u||_2^{n_2-2} \right) ||\triangle u||_2^2 + \int_{\Omega} k_\delta(x)|u_t|^{n(x)} dx \\
&\leq \delta \left( k_e^{n_1} \omega_0^{n_1} \left( \frac{2\mathcal{E}(0)}{\ell} \right)^{n_1-2} + k_e^{n_2} \omega_0^{n_2} \left( \frac{2\mathcal{E}(0)}{\ell} \right)^{n_2-2} \right) ||\triangle u||_2^2 + \int_{\Omega} k_\delta(x)|u_t|^{n(x)} dx \\
&\leq \delta k_0 ||\triangle u||_2^2 + \int_{\Omega} k_\delta(x)|u_t|^{n(x)} dx.
\end{aligned}
\tag{30}
$$

Combining (24)–(30) with (19) and fixing $\delta = \frac{\ell}{8k_0}$, we obtain (18). $\square$

**Lemma 5.** *Suppose that the assumptions* (C1) *and* (C2) *are satisfied; then, the functional*

$$
J(t) := \int_0^t f(t-s) \|\nabla u(s)\|_2^2 ds
$$

*satisfies the following estimate:*

$$
J'(t) \leq 3(1-\ell) \|\Delta u\|_2^2 - \tfrac{1}{2}(g \circ \nabla u)(t),
\tag{31}
$$

*where* $f(t) = \int_t^\infty g(s) ds$.

**Proof.** Applying Young's inequality, (C1) and the fact that $f(t) \leq f(0) = \frac{1-\ell}{\omega_0}$, we obtain, for any $t \geq 0$,

$$
\begin{aligned}
J'(t) &= f(0)\|\nabla u(t)\|_2^2 - \int_0^t g(t-s)\|\nabla u(s)\|_2^2 ds \\
&= f(t)\|\nabla u(t)\|_2^2 - \int_0^t g(t-s)\|\nabla(u(s) - u(t))\|_2^2 ds \\
&\quad - 2 \int_\Omega \int_0^t g(t-s)\nabla u(t).\nabla(u(s) - u(t)) ds dx \\
&\leq f(0)\|\nabla u(t)\|_2^2 - (g \circ \nabla u)(t) \\
&\quad + \frac{2}{\omega_0}(1-\ell)\|\nabla u(t)\|_2^2 + \frac{1}{2}(g \circ \nabla u)(t) \\
&= \frac{3}{\omega_0}(1-\ell)\|\nabla u(t)\|^2 - \frac{1}{2}(g \circ \nabla u)(t) \\
&\leq 3(1-\ell)\|\Delta u(t)\|_2^2 - \frac{1}{2}(g \circ \nabla u)(t).
\end{aligned}
$$

$\square$

**Lemma 6.** *Assume that* $n_1 \geq 2$. *Then, the functional* $\mathcal{L}$ *defined by*

$$
\mathcal{L}(t) := N\mathcal{E}(t) + \varepsilon_1 \mathbb{I}(t)
$$

*satisfies the following equivalence relation*

$$\mathcal{L} \sim \mathcal{E}, \tag{32}$$

*and for some positive constants N, $\varepsilon_1$, the functional satisfies the following estimate*

$$\mathcal{L}'(t) \leq -k\|u_t\|_2^2 - 4(1-\ell)\|\Delta u\|_2^2 + \tfrac{1}{4}(g \circ \nabla u)(t), \forall t \in [0, \infty). \tag{33}$$

**Proof.** The proof of (32) is completed in [22]. For the proof of (33), combining (14) and (17), and using the fact that $g'(t) := \varepsilon g(t) - h_\varepsilon(t)$ yields

$$\begin{aligned}
\mathcal{L}'(t) \leq &-\Big[N - \varepsilon_1 k_p - \frac{3\varepsilon_1 \omega_0}{\ell}\Big]\|\nabla u_t\|_2^2 - \frac{l}{4}\varepsilon_1\|\Delta u\|_2^2 + \frac{N\varepsilon}{2}(g \circ \nabla u)(t) \\
&-\Big[\frac{N}{2} - k\,K_\varepsilon\Big](h_\varepsilon \circ \Delta u)(t) - \int_\Omega [aN - k\varepsilon_1]|u_t|^{n(x)}dx.
\end{aligned} \tag{34}$$

First, we select $\varepsilon_1 = \frac{N}{2\left(k_\rho + \frac{3\omega_0}{\ell}\right)}$; then, we choose

$$N > \max\Big\{\varepsilon_1 k_p + \frac{3\varepsilon_1 \omega_0}{\ell}, \frac{k\varepsilon_1}{a}\Big\}.$$

Finally, we set $\varepsilon = \frac{1}{4N}$. Thus, (33) is established. $\square$

**Lemma 7.** *Assume that $n_1 \in (1, 2)$; then, the functional $\mathcal{L}$ defined by*

$$\mathcal{L}(t) := N\mathcal{E}(t) + \varepsilon_1 \mathbb{I}(t)$$

*satisfies*

$$\mathcal{L} \sim \mathcal{E}, \tag{35}$$

*for a suitable choice of the positive constants $N, \varepsilon_1$, the functional satisfies the following estimate*

$$\mathcal{L}'(t) \leq -k\|u_t\|_2^2 - 4(1-\ell)\|\Delta u\|_2^2 + \frac{1}{4}(g \circ \nabla u)(t) + k\big(-\mathcal{E}'(t)\big)^{n_1-1}, \forall t \in [0, \infty). \tag{36}$$

**Proof.** The proof is similar to the above arguments. $\square$

**Lemma 8.** *Assume that (C1) and (C2) hold; then, we have, for $n_1 \geq 2$,*

$$\int_0^\infty \mathcal{E}(s)ds < \infty. \tag{37}$$

**Proof.** Using Lemmas 5 and 7 we see that the functional $\mathcal{L}_1$ defined by

$$\mathcal{L}_1(t) := \mathcal{L}(t) + J(t),$$

satisfies, for any $t \in [0, \infty)$ and for some positive constant $k_1$,

$$\begin{aligned}
\mathcal{L}_1'(t) \leq &- k_1\|u_t\|_2^2 - (1-\ell)\|\Delta u\|_2^2 - \frac{1}{4}(g \circ \nabla u)(t) \\
\leq &- k\mathcal{E}(t).
\end{aligned}$$

We then conclude that

$$\int_0^\infty \mathcal{E}(s)ds < +\infty.$$

$\square$

**Lemma 9.** *Assume that (C1) and (C2) hold; then, for $1 < n_1 < 2$, we obtain*

$$\int_0^\infty \mathcal{E}^{\frac{1}{n_1-1}}(s)ds < +\infty. \tag{38}$$

*Moreover,*

$$\int_{t_0}^\infty \mathcal{E}(s)ds \leq k(t - t_0)^{2-n_1}, \qquad \forall t \geq t_0. \tag{39}$$

**Proof.** Using Lemmas (5) and (7), we obtain that the functional $\mathcal{L}_1$ defined by

$$\mathcal{L}_1(t) := \mathcal{L}(t) + J(t)$$

is non-negative and satisfies, for some $t \in [t_0, \infty)$ and for some positive constants $k_0$, $k_1$,

$$\begin{aligned}
\mathcal{L}_1'(t) &\leq -k_1\|u_t\|_2^2 - (1-\ell)\|\Delta u\|_2^2 - \frac{1}{4}(g \circ \nabla u)(t) + K\left[\int_\Omega |u_t|^n(x)dx\right]^{n_1-1} \\
&\leq -k_0\mathcal{E}(t) + k_1\big(-\mathcal{E}'(t)\big)^{n_1-1}.
\end{aligned} \tag{40}$$

Multiplying (40) by $\mathcal{E}^q(t)$, $q = \frac{2-n_1}{n_1-1} > 0$, and using Young's inequality, we obtain:

$$\begin{aligned}
\mathcal{E}^q(t)\mathcal{L}_1'(t) &\leq -k_0\mathcal{E}^{q+1}(t) + k_1\mathcal{E}^q(t)\big(-\mathcal{E}'(t)\big)^{n_1-1} \\
&\leq -k_0(1-\varepsilon)\mathcal{E}^{q+1}(t) + k(\varepsilon)\big(-\mathcal{E}'(t)\big).
\end{aligned} \tag{41}$$

Taking $\varepsilon$ to be small enough and the fact that $\mathcal{E}$ is non-increasing, (41) becomes:

$$\mathcal{E}^{q+1}(t) \leq -k\mathcal{L}_1'(t), \tag{42}$$

where $\mathcal{L}_1(t) = \mathcal{E}^q(t)\mathcal{L}_1(t) + k\mathcal{E}'(t)$. From this estimate, we conclude that

$$\int_0^\infty \mathcal{E}^{q+1}(s)ds < +\infty. \tag{43}$$

Since $q + 1 = \frac{1}{n_1-1}$, we end up with

$$\int_0^\infty \mathcal{E}^{\frac{1}{n_1-1}}(s)ds < +\infty. \tag{44}$$

Furthermore, from (44) and Hölder's inequality, we can deduce that

$$\int_{t_0}^t \mathcal{E}(s)ds \leq (t-t_0)^{\frac{q}{q+1}}\left[\int_{t_0}^t \mathcal{E}^{q+1}(s)ds\right]^{\frac{1}{q+1}} \leq k(t-t_0)^{\frac{q}{q+1}} = k(t-t_0)^{2-n_1}, \ \forall t \geq t_0. \tag{45}$$

$\square$

## 4. Decay Results

We begin with the statement of our main theorem.

**Theorem 1 (The case: $n_1 \geq 2$).** *Assume that (C1) and (C2) hold and the initial data $(u_0, u_1) \in H(\Omega) \times H_0^1(\Omega)$; then, there exist constants $\lambda_1, \lambda_2 \in (0, \infty)$, such that the energy functional $\mathcal{E}(t)$ associated to Problem (1)–(2) satisfies the estimate*

$$\mathcal{E}(t) \leq \lambda_2 \mathbb{G}_0^{-1}\left(\frac{\lambda_1}{\int_{t_0}^t \xi(s)ds}\right), \qquad \forall t > t_0, \tag{46}$$

*where $\mathbb{G}_0(\tau) = \tau\mathbb{G}'(\tau)$.*

**Proof.** Since $\xi$ is non-increasing, with the identity (14) and inequality (15),

$$
\int_0^{t_0} g(s) \|\nabla(u(t) - u(t-s))\|_2^2 ds \leq \frac{1}{\xi(t_0)} \int_0^{t_0} \xi(s) g(s) \|\nabla(u(t) - u(t-s))\|_2^2 ds
$$
$$
\leq -\frac{g(0)}{a\xi(t_0)} \int_0^{t_0} {}'(s) \|\nabla(u(t) - u(t-s))\|_2^2 ds \tag{47}
$$
$$
\leq -\tilde{k}\mathcal{E}'(t), \forall t \in [t_0, \infty).
$$

Combining (47) and (33), we obtain

$$
\mathcal{L}'(t) \leq -n\mathcal{E}(t) - k\mathcal{E}'(t) + k\int_{t_0}^t g(s) \|\nabla(u(t) - u(t-s))\|_2^2 ds, \forall t \in [t_0, \infty). \tag{48}
$$

Now, define a functional $\eta$ as

$$
\eta(t) := \gamma \int_{t_0}^t \|\nabla(u(t) - u(t-s))\|_2^2 ds, \quad \forall t \geq t_0.
$$

Using the inequality,

$$
\|\nabla u\|_2^2 \leq \omega_0 \|\Delta u\|_2^2, \ \forall u \in H(\Omega), \tag{49}
$$

we deduce that

$$
\mathcal{E}(t) \geq \frac{\ell}{2} \|\Delta u(t)\|_2^2 \quad \text{and} \quad \mathcal{E}(t) \geq \frac{\ell}{2\omega_0} \|\nabla u(t)\|_2^2, \quad \forall t \geq 0.
$$

These estimates and (37) yield

$$
\int_{t_0}^t \left( \|\nabla(u(t) - u(t-s))\|_2^2 \right) ds \leq 2\int_{t_0}^t \left( \|\nabla u(t)\|_2^2 + \|\nabla u(t-s)\|_2^2 \right) ds
$$
$$
\leq \frac{4\omega_0}{\ell} \int_{t_0}^t \left( \mathcal{E}(t) + \mathcal{E}(t-s) \right) ds
$$
$$
\leq \frac{8\omega_0}{\ell} \int_{t_0}^\infty \mathcal{E}(s) ds < +\infty, \quad \forall t \geq t_0.
$$

So, with $\gamma \in (0, 1)$, we obtain

$$
\eta(t) \in (1, \infty), \quad \forall t \in [t_0, \infty). \tag{50}
$$

Let $\theta$ be another functional defined by

$$
\theta(t) := -\int_{t_0}^t g'(s) \|\nabla(u(t) - u(t-s))\|_2^2 ds.
$$

In view of estimate (14), we can observe that $\mathcal{E}'(t) \leq \frac{1}{2}(g \circ \nabla u)(t)$. Therefore,

$$
\theta(t) \leq -k\mathcal{E}'(t), \quad \forall t \in [t_0, \infty). \tag{51}
$$

Next, the facts that $\mathbb{G}$ is strictly convex and $\mathbb{G}(0) = 0$ give that

$$
\mathbb{G}(s\tau) \leq s\mathbb{G}(\tau), \quad \text{for} \quad s \in [0, 1] \quad \text{and} \quad \tau \in (0, r].
$$

In view of the assumptions (C1), (C2), (50) and Jensen's inequality, we obtain

$$
\begin{aligned}
\theta(t) &= -\frac{1}{\eta(t)} \int_{t_0}^{t} \eta(t)(s) \|\nabla(u(t) - u(t-s))\|_2^2 ds \\
&\geq \frac{1}{\eta(t)} \int_{t_0}^{t} \eta(t)\xi(s)\mathbb{G}(g(s))\|\nabla(u(t) - u(t-s))\|_2^2 ds \\
&\geq \frac{\xi(t)}{\eta(t)} \int_{t_0}^{t} \mathbb{G}(\eta(t)g(s))\|\nabla(u(t) - u(t-s))\|_2^2 ds \\
&\geq \frac{\xi(t)}{\gamma} \mathbb{G}\left(\gamma \int_{t_0}^{t} g(s)\|\nabla(u(t) - u(t-s))\|_2^2 ds\right) \\
&\geq \frac{\xi(t)}{\gamma} \mathbb{G}\left(\gamma \int_{t_0}^{t} g(s)\|\nabla(u(t) - u(t-s))\|_2^2 ds\right), \forall t \in [t_0, \infty).
\end{aligned}
$$

This yields, for any $t \geq t_0$,

$$
\int_{t_0}^{t} g(s)\|\nabla(u(t) - u(t-s))\|_2^2 ds \leq \frac{1}{\gamma}\bar{\mathbb{G}}^{-1}\left(\frac{\gamma\theta(t)}{\xi(t)}\right).
$$

Therefore, (48) becomes

$$
\mathcal{F}'(t) \leq -n\mathcal{E}(t) + \frac{k}{\gamma}\bar{\mathbb{G}}^{-1}\left(\frac{\gamma\theta(t)}{\xi(t)}\right), \qquad \forall t \in [t_0, \infty) \tag{52}
$$

where $\mathcal{F} := \mathcal{L} + kE$.

Let $r_1 \in (0, r)$, and define a functional $\mathcal{F}_1$ by

$$
\mathcal{F}_1(t) := \bar{\mathbb{G}}'\left(\frac{r_1\mathcal{E}(t)}{\mathcal{E}(0)}\right)\mathcal{F}(t), \qquad \forall t \geq t_0.
$$

Then, using the facts that $\mathcal{E}' \leq 0$, $\mathbb{G}' > 0$ and $\mathbb{G}'' > 0$ together with estimate (52) lead to

$$
\begin{aligned}
\mathcal{F}_1'(t) &= \frac{r_1\mathcal{E}'(t)}{\mathcal{E}(0)}\bar{\mathbb{G}}''\left(\frac{r_1\mathcal{E}(t)}{\mathcal{E}(0)}\right)\mathcal{F}(t) + \bar{\mathbb{G}}'\left(\frac{r_1\mathcal{E}(t)}{\mathcal{E}(0)}\right)\mathcal{F}'(t) \\
&\leq -n\mathcal{E}(t)\bar{\mathbb{G}}'\left(\frac{r_1\mathcal{E}(t)}{\mathcal{E}(0)}\right) + \frac{k}{\gamma}\bar{\mathbb{G}}'\left(\frac{r_1\mathcal{E}(t)}{\mathcal{E}(0)}\right)\bar{\mathbb{G}}^{-1}\left(\frac{\gamma\theta(t)}{\xi(t)}\right), \quad \forall t \geq t_0. \tag{53}
\end{aligned}
$$

$\mathbb{G}^*$ is the convex conjugate of $\bar{\mathbb{G}}$ (see ([23], pp. 61–64)), that is

$$
\mathbb{G}^*(s) = s(\bar{\mathbb{G}}')^{-1}(s) - \bar{\mathbb{G}}\left[(\bar{\mathbb{G}}')^{-1}(s)\right] \tag{54}
$$

and satisfies the generalized Young inequality

$$
AB \leq \bar{\mathbb{G}}^*(A) + \bar{\mathbb{G}}(B). \tag{55}
$$

Now, set

$$
A = \bar{\mathbb{G}}'\left(\frac{r_1\mathcal{E}(t)}{\mathcal{E}(0)}\right) \quad \text{and} \quad B = \bar{\mathbb{G}}^{-1}\left(\frac{\gamma\theta(t)}{\xi(t)}\right),
$$

then, in view of (53) and (55), we arrive at

$$
\begin{aligned}
\mathcal{F}_1'(t) &\leq -n\mathcal{E}(t)\bar{\mathbb{G}}'\left(\frac{r_1\mathcal{E}(t)}{\mathcal{E}(0)}\right) + \frac{k}{\gamma}\bar{\mathbb{G}}^*\left[\bar{\mathbb{G}}'\left(\frac{r_1\mathcal{E}(t)}{\mathcal{E}(0)}\right)\right] + \frac{k\theta(t)}{\xi(t)} \\
&\leq -n(\mathcal{E}(0) - kr_1)\frac{\mathcal{E}(t)}{\mathcal{E}(0)}\bar{\mathbb{G}}'\left(\frac{r_1\mathcal{E}(t)}{\mathcal{E}(0)}\right) + k\frac{\theta(t)}{\xi(t)}, \qquad \forall t \geq t_0.
\end{aligned}
$$

Fixing $r_1$, we obtain

$$\mathcal{F}_1'(t) \leq -n_1 \frac{\mathcal{E}(t)}{\mathcal{E}(0)} \bar{\mathbb{G}}'\left(\frac{r_1\mathcal{E}(t)}{\mathcal{E}(0)}\right) + k\frac{\theta(t)}{\xi(t)}, \qquad \forall t \geq t_0, \tag{56}$$

where $n_1 > 0$. Multiplying both sides of (56) by $\xi(t)$ $r_1\dfrac{\mathcal{E}(t)}{\mathcal{E}(0)} < r$ and using inequality (51), yields

$$\begin{aligned}
\xi(t)\mathcal{F}_1'(t) &\leq -n_1\frac{\mathcal{E}(t)}{\mathcal{E}(0)}\mathbb{G}'\left(\frac{r_1\mathcal{E}(t)}{\mathcal{E}(0)}\right)\xi(t) + k\theta(t)\\
&\leq -n_1\frac{\mathcal{E}(t)}{\mathcal{E}(0)}\mathbb{G}'\left(\frac{r_1\mathcal{E}(t)}{\mathcal{E}(0)}\right)\xi(t) - k\mathcal{E}'(t), \qquad \forall t \geq t_0.
\end{aligned}$$

Let $\mathcal{F}_2 = \xi\mathcal{F}_1 + k\mathcal{E}$; then, we obtain from the non-increasing property of $\xi$ that

$$n_1\frac{\mathcal{E}(t)}{\mathcal{E}(0)}\mathbb{G}'\left(\frac{r_1\mathcal{E}(t)}{\mathcal{E}(0)}\right)\xi(t) \leq -\mathcal{F}_2'(t), \qquad \forall t \geq t_0. \tag{57}$$

The map

$$t \longmapsto \mathcal{E}(t)\mathbb{G}'\left(\frac{\varepsilon_1\mathcal{E}(t)}{\mathcal{E}(0)}\right)$$

is non-increasing, since $\mathbb{G}'' > 0$ and $\mathcal{E}$ non-increasing. As a result, integrating (57) over $(t_0, t)$ yields

$$\begin{aligned}
n_1\frac{\mathcal{E}(t)}{\mathcal{E}(0)}\mathbb{G}'\left(\frac{r_1\mathcal{E}(t)}{\mathcal{E}(0)}\right)\int_{t_0}^t \xi(s)ds &\leq \int_{t_0}^t \frac{\mathcal{E}(s)}{\mathcal{E}(0)\mathbb{G}}{}'\left(\frac{r_1\mathcal{E}(s)}{\mathcal{E}(0)}\right)\xi(s)ds\\
&\leq \mathcal{F}_2(t_0) - \mathcal{F}_2(t)\\
&\leq \mathcal{F}_2(t_0), \qquad \forall t \geq t_0.
\end{aligned}$$

Finally, we set $\mathbb{G}_0(\tau) = \tau\mathbb{G}'(\tau)$. Then, we obtain for some positive constants $\lambda_1$ and $\lambda_2$ the following estimate

$$\mathcal{E}(t) \leq \lambda_2\mathbb{G}_0^{-1}\left(\frac{\lambda_1}{\int_{t_0}^t \xi(s)ds}\right), \qquad \forall t > t_0.$$

$\square$

We consider the following examples:

**Example 1.** *(1) Take $g(t) = \lambda e^{-\beta t}$, $t \in [0, \infty]$, $\lambda, \beta \in (0, \infty)$ as constants. The constant $\lambda$ is carefully chosen to satisfy the assumption (C1). Consequently,*

$$g'(t) = -\beta\mathbb{G}(g(t)), \qquad \xi(t) = \beta \qquad \text{and} \qquad \mathbb{G}(s) = s.$$

*Hence, in view of Theorem 1, we conclude that for some constant $K, t_0 \in (0, \infty)$,*

$$\mathcal{E}(t) \leq \frac{K}{t - t_0}, \qquad \forall t \in [t_0, \infty).$$

*(2) Let $g(t) = \lambda e^{-(1+t)^\gamma}$, for $t \in [0, \infty)$, $\gamma \in (0, 1)$ and $\lambda$ selected such that (C1) is satisfied. Then,*

$$g'(t) = -\xi(t)\mathbb{G}(g(t)), \qquad \xi(t) = \gamma(1+t)^{\gamma-1} \qquad \text{and} \qquad \mathbb{G}(s) = s.$$

*In view of Theorem 1, we deduce that for some constant* $K, t_0 \in (0, \infty)$,

$$\mathcal{E}(t) \leq \frac{K}{(1+t)^\gamma}, \qquad \text{for} \quad t \quad \text{large enough.}$$

(3)  *For* $\gamma \in (1, \infty)$, *let*

$$g(t) = \frac{\lambda}{(1+t)^\gamma}, \qquad t \in [0, \infty)$$

*and* $\lambda$ *carefully chosen so that* (C1) *is valid. Then,*

$$g'(t) = -\beta \mathbb{G}(g(t)), \qquad \xi(t) = \beta \qquad \text{and} \qquad \mathbb{G}(s) = s^p,$$

*with* $p = \frac{1+\gamma}{\gamma} \in (1,2)$, *and* $\beta$ *is a positive constant. It follows from Theorem 1 that for some constants* $K, t_0 \in (0, \infty)$,

$$\mathcal{E}(t) \leq \frac{K}{(1+t)^{\gamma/(\gamma+1)}}, \qquad \forall t > t_0.$$

**Theorem 2 (The case: $1 < n_1 < 2$).** *Assume that hypotheses* (C1) *and* (C2) *hold and the data* $(u_0, u_1) \in H(\Omega) \times H_0^1(\Omega)$. *Then, there exist positive constants* $\lambda_1$, $\lambda_2$ *such that the energy functional associated to Problem* (1)–(2) *satisfies the estimate*

$$\mathcal{E}(t) \leq \lambda_2 (t-t_0)^{2-n_1} \mathbb{G}_0^{-1} \left( \frac{\lambda_1}{(t-t_0)^{\frac{2-n_1}{n_1-1}} \int_{t_0}^t \xi(s) ds} \right), \qquad \forall t > t_0, \tag{58}$$

*where* $\mathbb{G}_0(\tau) = \tau \mathbb{G}'(\tau)$.

**Proof.** Similar to the proof of Theorem 1, we use (47) and (33) to obtain

$$\mathcal{L}'(t) \leq -n\mathcal{E}(t) - k\mathcal{E}'(t) + k \int_{t_0}^t g(s) \|\nabla(u(t) - u(t-s))\|_2^2 ds + k \left[ -\mathcal{E}'(t) \right]^{n_1-1}, \forall t \in [t_0, \infty). \tag{59}$$

We then define another functional $\eta$ as

$$\eta(t) := \frac{\gamma}{(t-t_0)^{2-n_1}} \int_{t_0}^t \left( \|\nabla(u(t) - u(t-s))\|_2^2 \right) ds, \quad \forall t \geq t_0.$$

Using (39), we conclude that

$$\eta(t) \leq k\gamma, \tag{60}$$

then choosing $0 < \gamma < 1$ small enough so that

$$\eta(t) < 1, \qquad \forall t \geq t_0. \tag{61}$$

Combining this with the hypotheses (C1), (C2), Jensen's inequality and (61), we obtain

$$
\begin{aligned}
\theta(t) &= -\frac{1}{\eta(t)} \int_{t_0}^{t} \eta(t)(s) \|\nabla(u(t) - u(t-s))\|_2^2 ds \\
&\geq \frac{1}{\eta(t)} \int_{t_0}^{t} \eta(t)\xi(s)\mathbb{G}(g(s)) \|\nabla(u(t) - u(t-s))\|_2^2 ds \\
&\geq \frac{\xi(t)}{\eta(t)} \int_{t_0}^{t} \mathbb{G}(\eta(t)g(s)) \|\nabla(u(t) - u(t-s))\|_2^2 ds \\
&\geq \frac{(t-t_0)^{2-n_1}\xi(t)}{\gamma} \mathbb{G}\left( \frac{\gamma}{(t-t_0)^{2-n_1}} \int_{t_0}^{t} g(s) \|\nabla(u(t) - u(t-s))\|_2^2 ds \right) \\
&\geq \frac{(t-t_0)^{2-n_1}\xi(t)}{\gamma} \bar{\mathbb{G}}\left( \frac{\gamma}{(t-t_0)^{2-n_1}} \int_{t_0}^{t} g(s) \|\nabla(u(t) - u(t-s))\|_2^2 ds \right), \forall t \in (t_0, \infty),
\end{aligned}
$$

where $\bar{\mathbb{G}}$ is a $C^2$ extension of $\mathbb{G}$ which is strictly increasing and strictly convex on $(0, \infty)$. This yields, for any $t \geq t_0$,

$$
\int_{t_0}^{t} g(s) \|\nabla(u(t) - u(t-s))\|_2^2 ds \leq \frac{(t-t_0)^{2-n_1}}{\gamma} \bar{\mathbb{G}}^{-1}\left( \frac{\gamma\theta(t)}{(t-t_0)^{2-n_1}\xi(t)} \right) \tag{62}
$$

and (59) becomes

$$
\mathcal{F}'(t) \leq -n\mathcal{E}(t) + \frac{(t-t_0)^{2-n_1}}{\gamma} \bar{\mathbb{G}}^{-1}\left( \frac{\gamma\theta(t)}{(t-t_0)^{2-n_1}\xi(t)} \right) + \left[ -\mathcal{E}'(t) \right]^{n_1-1}, \qquad \forall t \geq t_0, \tag{63}
$$

where $\mathcal{F} := \mathcal{L} + k\mathcal{E}$. Let $0 < r_1 < r$; then, define a functional $\mathcal{F}_1$ by

$$
\mathcal{F}_1(t) := \bar{\mathbb{G}}'\left( \frac{r_1}{(t-t_0)^{2-n_1}} \cdot \frac{\mathcal{E}(t)}{\mathcal{E}(0)} \right) \mathcal{F}(t), \qquad \forall t \geq t_0.
$$

Since $\mathcal{E}' \leq 0$, $\mathbb{G}' > 0$ and $\mathbb{G}'' > 0$, we obtain

$$
\begin{aligned}
\mathcal{F}_1'(t) &= \left[ \frac{(n-2)r_1}{n(t-t_0)^{\frac{n+2}{n}}} \frac{\mathcal{E}(t)}{\mathcal{E}(0)} + \frac{r_1}{(t-t_0)^{2-n_1}} \frac{\mathcal{E}'(t)}{\mathcal{E}(0)} \right] \bar{\mathbb{G}}''\left( \frac{r_1}{(t-t_0)^{2-n_1}} \frac{\mathcal{E}(t)}{\mathcal{E}(0)} \right) \mathcal{F}(t) \\
&\quad + \bar{\mathbb{G}}'\left( \frac{r_1}{(t-t_0)^{2-n_1}} \frac{\mathcal{E}(t)}{\mathcal{E}(0)} \right) \mathcal{F}'(t) \\
&\leq \bar{\mathbb{G}}'\left( \frac{r_1}{(t-t_0)^{2-n_1}} \cdot \frac{\mathcal{E}(t)}{\mathcal{E}(0)} \right) \mathcal{F}'(t), \quad \forall t \geq t_0. \tag{64}
\end{aligned}
$$

Estimates (63) and (64) imply that

$$
\begin{aligned}
\mathcal{F}_1'(t) &\leq -n\mathcal{E}(t)\bar{\mathbb{G}}'\left( \frac{r_1}{(t-t_0)^{2-n_1}} \cdot \frac{\mathcal{E}(t)}{\mathcal{E}(0)} \right) + \frac{(t-t_0)^{2-n_1}}{\gamma} \bar{\mathbb{G}}^{-1}\left( \frac{\gamma\theta(t)}{(t-t_0)^{2-n_1}\xi(t)} \right) \bar{\mathbb{G}}'\left( \frac{r_1}{(t-t_0)^{2-n_1}} \cdot \frac{\mathcal{E}(t)}{\mathcal{E}(0)} \right) \\
&\quad + c\bar{\mathbb{G}}'\left( \frac{r_1}{(t-t_0)^{2-n_1}} \cdot \frac{\mathcal{E}(t)}{\mathcal{E}(0)} \right) \left[ -\mathcal{E}'(t) \right]^{n_1-1}, \quad \forall t \geq t_0. \tag{65}
\end{aligned}
$$

Let $\bar{\mathbb{G}}^*$ be defined as in (54) and satisfy (55). Set

$$
A = \bar{\mathbb{G}}'\left( \frac{r_1}{(t-t_0)^{2-n_1}} \cdot \frac{\mathcal{E}(t)}{\mathcal{E}(0)} \right) \qquad \text{and} \qquad B = \bar{\mathbb{G}}^{-1}\left( \frac{\gamma\theta(t)}{(t-t_0)^{2-n_1}\xi(t)} \right);
$$

then, it follows from a combination of (55) and (64) that

$$
\begin{aligned}
\mathcal{F}_1'(t) &\leq -n\mathcal{E}(t)\bar{\mathbb{G}}'\left( \frac{r_1}{(t-t_0)^{2-n_1}} \cdot \frac{\mathcal{E}(t)}{\mathcal{E}(0)} \right) + \frac{k(t-t_0)^{2-n_1}}{\gamma} \bar{\mathbb{G}}^*\left[ \bar{\mathbb{G}}'\left( \frac{r_1}{(t-t_0)^{2-n_1}} \cdot \frac{\mathcal{E}(t)}{\mathcal{E}(0)} \right) \right] + \frac{\theta(t)}{\xi(t)} \\
&\quad + k\bar{\mathbb{G}}'\left( \frac{r_1}{(t-t_0)^{2-n_1}} \cdot \frac{\mathcal{E}(t)}{\mathcal{E}(0)} \right) \left[ -\mathcal{E}'(t) \right]^{n_1-1}.
\end{aligned}
$$

By definition of $\bar{\mathbb{G}}^*$ and since $\bar{\mathbb{G}} > 0$, we have:

$$\mathcal{F}_1'(t) \leq -n\mathcal{E}(t)\bar{\mathbb{G}}'\left(\frac{r_1}{(t-t_0)^{2-n_1}} \cdot \frac{\mathcal{E}(t)}{\mathcal{E}(0)}\right) + k\, r_1 \frac{\mathcal{E}(t)}{\mathcal{E}(0)}\bar{\mathbb{G}}'\left(\frac{r_1}{(t-t_0)^{2-n_1}} \cdot \frac{\mathcal{E}(t)}{\mathcal{E}(0)}\right) + \frac{\theta(t)}{\bar{\xi}(t)}$$
$$+ k\bar{\mathbb{G}}'\left(\frac{r_1}{(t-t_0)^{2-n_1}} \cdot \frac{\mathcal{E}(t)}{\mathcal{E}(0)}\right)\left[-\mathcal{E}'(t)\right]^{n_1-1},$$

and it can be written as

$$\mathcal{F}_1'(t) \leq -n(\mathcal{E}(0) - kr_1)\frac{\mathcal{E}(t)}{\mathcal{E}(0)}\bar{\mathbb{G}}'\left(\frac{r_1}{(t-t_0)^{2-n_1}} \cdot \frac{\mathcal{E}(t)}{\mathcal{E}(0)}\right) + \frac{\theta(t)}{\bar{\xi}(t)} + k\bar{\mathbb{G}}'\left(\frac{r_1}{(t-t_0)^{2-n_1}} \cdot \frac{\mathcal{E}(t)}{\mathcal{E}(0)}\right)\left[-\mathcal{E}'(t)\right]^{n_1-1}.$$

After fixing $r_1$, we arrive at

$$\mathcal{F}_1'(t) \leq -n_1 \frac{\mathcal{E}(t)}{\mathcal{E}(0)}\bar{\mathbb{G}}'\left(\frac{r_1}{(t-t_0)^{2-n_1}} \cdot \frac{\mathcal{E}(t)}{\mathcal{E}(0)}\right) + \frac{\theta(t)}{\bar{\xi}(t)} + k\bar{\mathbb{G}}'\left(\frac{r_1}{(t-t_0)^{2-n_1}} \cdot \frac{\mathcal{E}(t)}{\mathcal{E}(0)}\right)\left[-\mathcal{E}'(t)\right]^{n_1-1}, \quad (66)$$

where $n_1 > 0$. Multiplying both sides of (66) by $\xi(t)\mathcal{E}^{\frac{2-n_1}{n_1-1}}(t)$, we reach

$$\xi(t)\mathcal{E}^{\frac{2-n_1}{n_1-1}}(t)\mathcal{F}_1'(t) \leq -n_1 \frac{\mathcal{E}(t)}{\mathcal{E}(0)}\xi(t)\mathcal{E}^{\frac{2-n_1}{n_1-1}}(t)\bar{\mathbb{G}}'\left(\frac{r_1}{(t-t_0)^{2-n_1}} \cdot \frac{\mathcal{E}(t)}{\mathcal{E}(0)}\right) - k_1(\mathcal{E}'(t))\mathcal{E}^{\frac{2-n_1}{n_1-1}}(t)$$
$$+ k\xi(t)\bar{\mathbb{G}}'\left(\frac{r_1}{(t-t_0)^{2-n_1}} \cdot \frac{\mathcal{E}(t)}{\mathcal{E}(0)}\right)\mathcal{E}^{\frac{2-n_1}{n_1-1}}(t)(-\mathcal{E}'(t))^{n_1-1}.$$

Using Young's inequality with $p = \frac{1}{n_1-1}$ and $q = \frac{1}{2-n_1}$, we obtain

$$\xi(t)\mathcal{E}^{\frac{2-n_1}{n_1-1}}(t)\mathcal{F}_1'(t) \leq (k\varepsilon - \frac{n_1}{\mathcal{E}(0)})\xi(t)\mathcal{E}^{\frac{2-n_1}{n_1-1}}(t)\bar{\mathbb{G}}'\left(\frac{r_1}{(t-t_0)^{2-n_1}} \cdot \frac{\mathcal{E}(t)}{\mathcal{E}(0)}\right)$$
$$- k_1(\mathcal{E}'(t) + \mathcal{E}^{\frac{2-n_1}{n_1-1}}(t)) - k_2\mathcal{E}'(t). \quad (67)$$

Let $\mathcal{F}_2 = \xi\mathcal{F}_1\mathcal{E}^{\frac{2-n}{2n-2}} + k_1\mathcal{E}\mathcal{E}^{\frac{2-n}{2n-2}} + k_2\mathcal{E}(t)$; then, we obtain, from the non-increasing property of $\xi$ and the fact that $\mathcal{E}' \leq 0$ and for $\varepsilon$ small enough that

$$\mathcal{F}_2'(t) \leq -n_2\xi(t)\mathcal{E}^{\frac{2-n_1}{n_1-1}}(t)\bar{\mathbb{G}}'\left(\frac{r_1}{(t-t_0)^{2-n_1}} \cdot \frac{\mathcal{E}(t)}{\mathcal{E}(0)}\right), \quad (68)$$

for some $n_2 > 0$. Then, we have for $n_3 = n_2\mathcal{E}(0)$ and $\frac{r_1}{(t-t_0)^{2-n_1}} \cdot \frac{\mathcal{E}(t)}{\mathcal{E}(0)} < r$ for small $r_1$,

$$n_3\left(\frac{\mathcal{E}^{\frac{2-n_1}{n_1-1}}(t)}{\mathcal{E}(0)}\right)\mathbb{G}'\left(\frac{r_1}{(t-t_0)^{2-n_1}} \cdot \frac{\mathcal{E}(t)}{\mathcal{E}(0)}\right)\xi(t) \leq -\mathcal{F}_2'(t). \quad (69)$$

Integrating (69) over $(t_0, t)$ yields

$$\int_{t_0}^{t} n_3 \frac{\mathcal{E}^{\frac{2-n_1}{n_1-1}}(s)}{\mathcal{E}(0)}\mathbb{G}'\left(\frac{r_1}{(s-t_0)^{2-n_1}} \cdot \frac{\mathcal{E}(s)}{\mathcal{E}(0)}\right)\xi(s)ds \leq -\int_{t_0}^{t}\mathcal{F}_2'(s)ds \leq \mathcal{F}_2'(t_0). \quad (70)$$

It follows from the fact that $\mathbb{G}'' > 0$ and non-increasing property of $\mathcal{E}(t)$ that the map

$$t \longmapsto \frac{\mathcal{E}^{\frac{2-n_1}{n_1-1}}(s)}{\mathcal{E}(0)}\left(\frac{r_1}{(s-t_0)^{2-n_1}} \cdot \frac{\mathcal{E}(s)}{\mathcal{E}(0)}\right)$$

is non-increasing. Consequently, we have

$$n_3\frac{\mathcal{E}^{\frac{2-n_1}{n_1-1}}(t)}{\mathcal{E}(0)}\mathbb{G}'\left(\frac{r_1}{(t-t_0)^{2-n_1}} \cdot \frac{\mathcal{E}(t)}{\mathcal{E}(0)}\right)\int_{t_0}^{t}\xi(s)ds \leq \int_{t_0}^{t} n_3\frac{\mathcal{E}^{\frac{2-n_1}{n_1-1}}(s)}{\mathcal{E}(0)}\mathbb{G}'\left(\frac{r_1}{(s-t_0)^{2-n_1}} \cdot \frac{\mathcal{E}(s)}{\mathcal{E}(0)}\right)\xi(s)ds$$
$$\leq -\int_{t_0}^{t}\mathcal{F}_2'(s)ds \leq \mathcal{F}_2'(t_0) = n_4. \quad (71)$$

To finish the proof of Theorem 2, we multiply (71) by $\left(\frac{1}{(t-t_0)}\right)^{\frac{2-n_1}{n_1-1}}$ to obtain

$$n_5 \left[\frac{\mathcal{E}(t)}{\mathcal{E}(0)(t-t_0)}\right]^{\frac{2-n_1}{n_1-1}} \mathbb{G}'\left(\frac{r_1}{(t-t_0)^{2-n_1}} \cdot \frac{\mathcal{E}(t)}{\mathcal{E}(0)}\right) \int_{t_0}^{t} \xi(s)ds \leq \left(\frac{n_4}{t-t_0}\right)^{\frac{2-n_1}{n_1-1}}. \tag{72}$$

Next, we set $\mathbb{G}_0(\tau) = \tau^{\frac{1}{n_1-1}} \mathbb{G}'(\tau)$ which is strictly increasing; then, we obtain for two positive constants $\lambda_1$ and $\lambda_2$

$$\mathcal{E}(t) \leq \lambda_2(t-t_0)^{2-n_1}\mathbb{G}_0^{-1}\left(\frac{\lambda_1}{(t-t_0)^{\frac{2-n_1}{n_1-1}}\int_{t_0}^{t}\xi(s)ds}\right), \qquad \forall\, t > t_0.$$

□

**Example 2.** *(1)   Let* $g(t) = \lambda e^{-\beta(1+t)^\gamma}$, $t \in [0,\infty)$, $\lambda, \beta \in (0,\infty)$ *and* $\gamma \in (0,1)$ *and* $\lambda$ *is selected such that* (C1) *is satisfied; then,* $g'(t) = -\beta\mathbb{G}(g(t))$ *with* $\xi(t) = \gamma(1+t)^{\gamma-1}$ *and* $\mathbb{G}(s) = s$. *So, in view of Theorem 2, we conclude that the solution of* (1)–(2) *satisfies the energy estimate*

$$\mathcal{E}(t) \leq \frac{K}{(t-t_0)^{n_1-1}}, \qquad \forall\, t > t_1.$$

*(2)   Suppose*

$$g(t) = \lambda(1+t)^{-\gamma}, \qquad \gamma \in (1,\infty),$$

*and* $\lambda$ *is chosen so that* (C1) *is satisfied. Then, for a positive constant* $\beta$,

$$g'(t) = -\beta\mathbb{G}(g(t)), \qquad \xi(t) = \beta \qquad \text{and} \qquad \mathbb{G}(s) = s^p, \qquad \forall\, p = \frac{1+\gamma}{\gamma}.$$

*Similar to the arguments in Example 1, we deduce that the solution of* (1)–(2) *satisfies the energy estimate*

$$\mathcal{E}(t) \leq \frac{K}{(t-t_0)^\mu},$$

*where* $\mu = \frac{(n_1-1)(n_1+\gamma-2)}{n_1+\gamma-1} \in (0,\infty)$, *and for sufficiently large t and some constant* $K, t_0 \in (0,\infty)$.

## 5. Numerical Results

We produce some numerical experiments in this section to demonstrate the theoretical findings of Theorems 1 and 2. For this reason, we discretize our Problem (1) in the time–space domain $(0,1] \times [0,1]$ using the second-order finite difference method (FDM) in time and fourth-order in space. The time interval $(0,T)$ is split into $N = 10{,}000$ subintervals with a time step $\Delta t = \frac{T}{N}$, and the spatial interval $(0,1)$ is divided into 50 subintervals. The homogeneous Dirichlet boundary condition for Problem (1) is stated, and $\Delta u = 0$ at boundary. Based on the relaxation function $\mathbb{G}$ and the initial conditions $u(x,0) = sin(\pi x)$ and $u_t(x,0) = 0$, we contrast the following numerical two tests.

- **Test 1:** We show the exponential decay of the solution of the problem (1) and the energy function given by (13), using $g(t) = e^{-t}$ and $n(x) = 2 + \frac{1}{x+1}$.

- **Test 2:** In the second test, we let $g(t) = \frac{1}{(t+1)^2}$ and $n(x) = 3 + \frac{1}{x+1}$.

In Figures 1 and 2, we show the cross-sections of the approximate solution $u$ at $x = 0.3$, $x = 0.5$, $x = 0.6$, and $x = 0.7$ for Test 1 and Test 2, respectively. In Figures 3 and 4, we graph

the corresponding energy functional (13). In addition, we show the decay behavior of the whole wave over the time interval $[0, 1]$ in Figures 5 and 6 for Test 1 and Test 2, respectively.

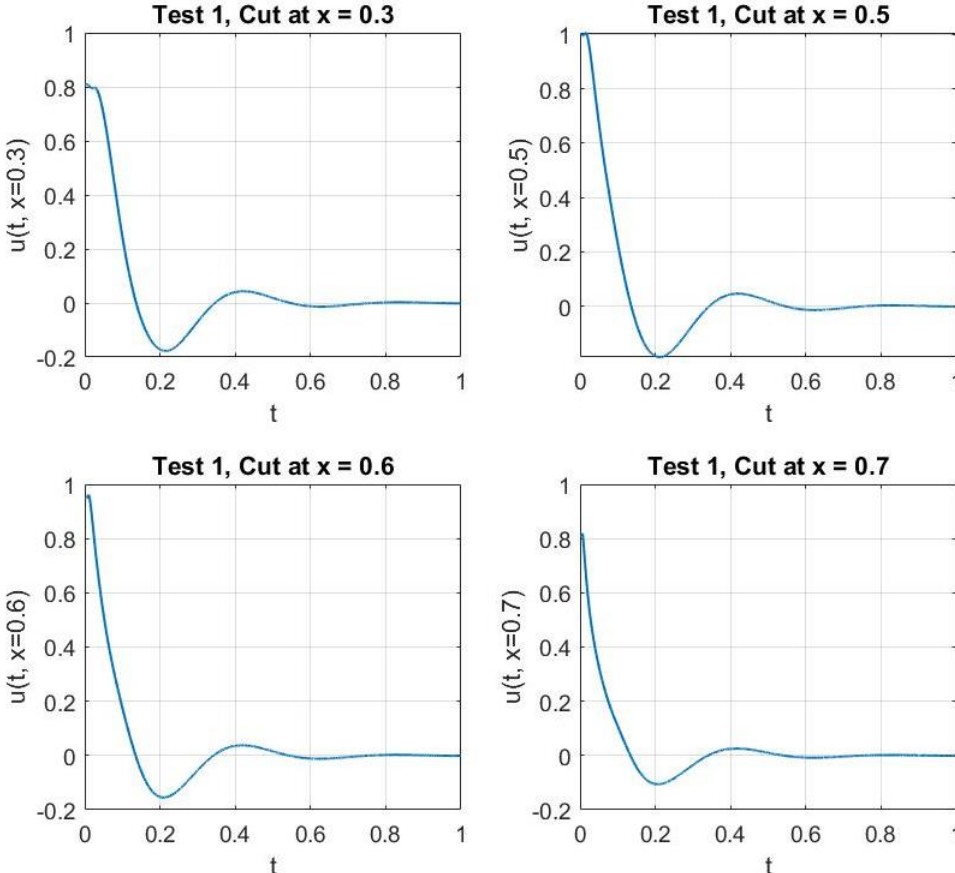

**Figure 1.** Test 1: The solution $u(t)$ of the problem at fixed values of $x$.

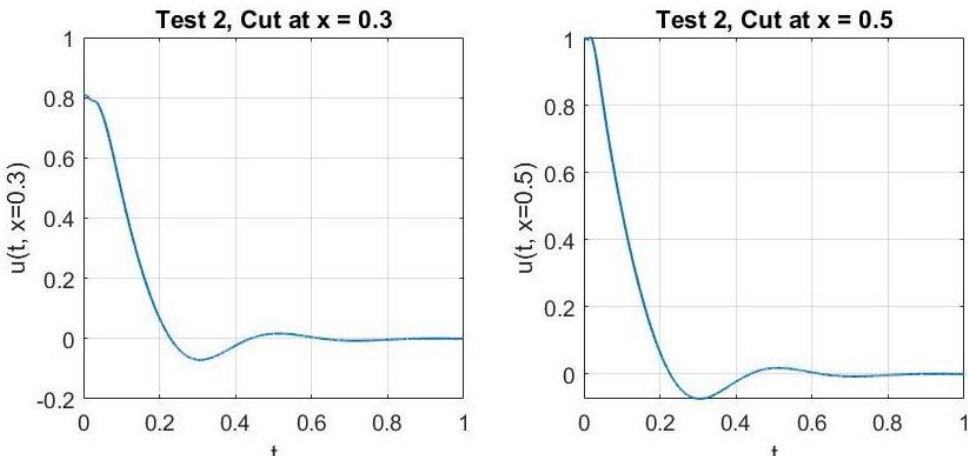

**Figure 2.** *Cont.*

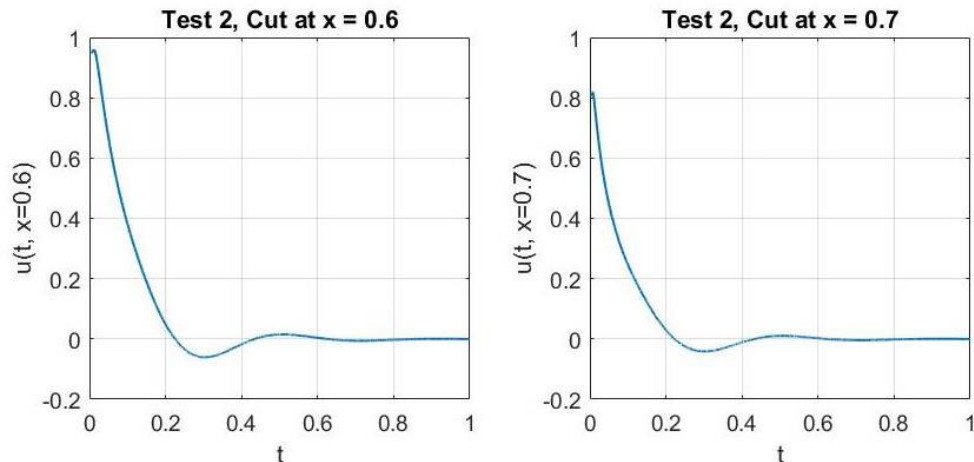

**Figure 2.** Test 2: The solution $u(t)$ of the problem at fixed values of $x$.

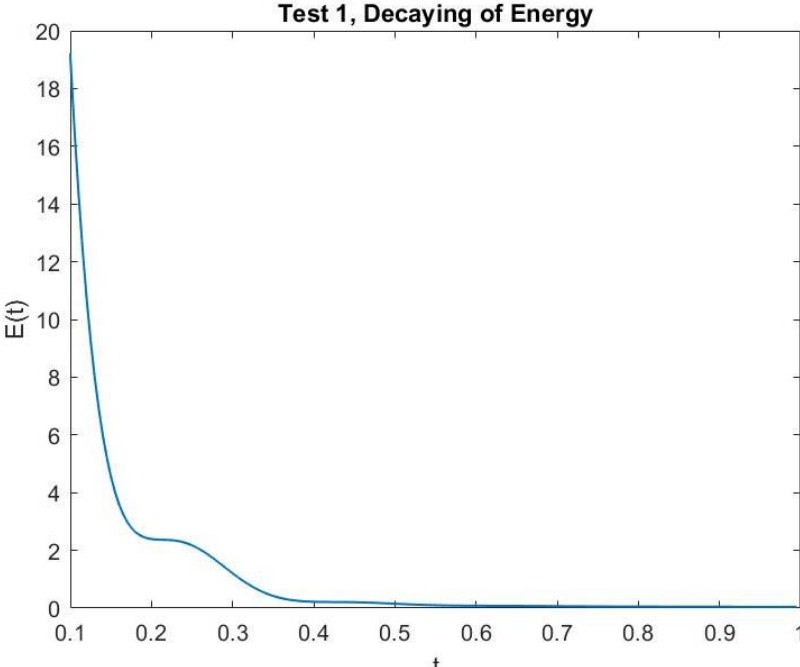

**Figure 3.** Test 1: The energy decay.

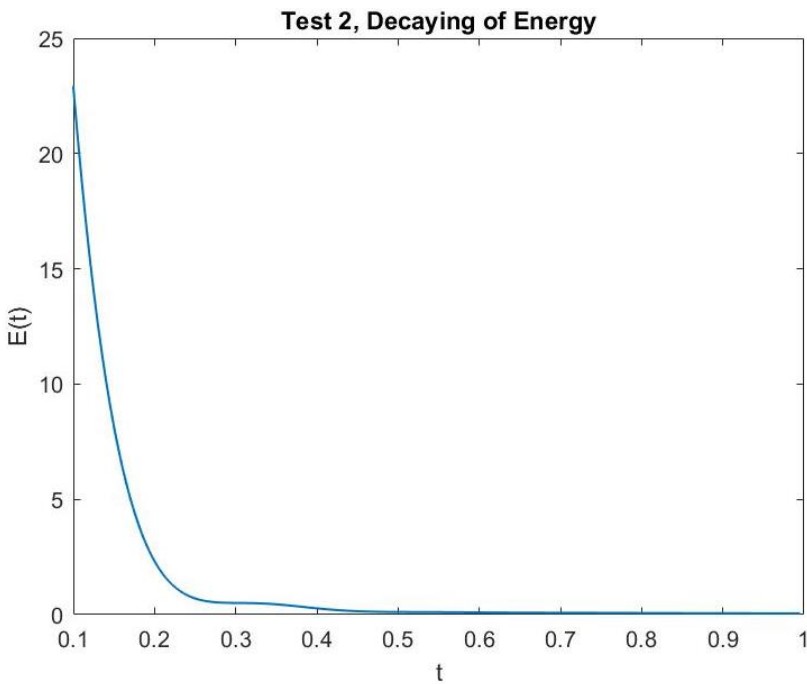

**Figure 4.** Test 2: The energy decay.

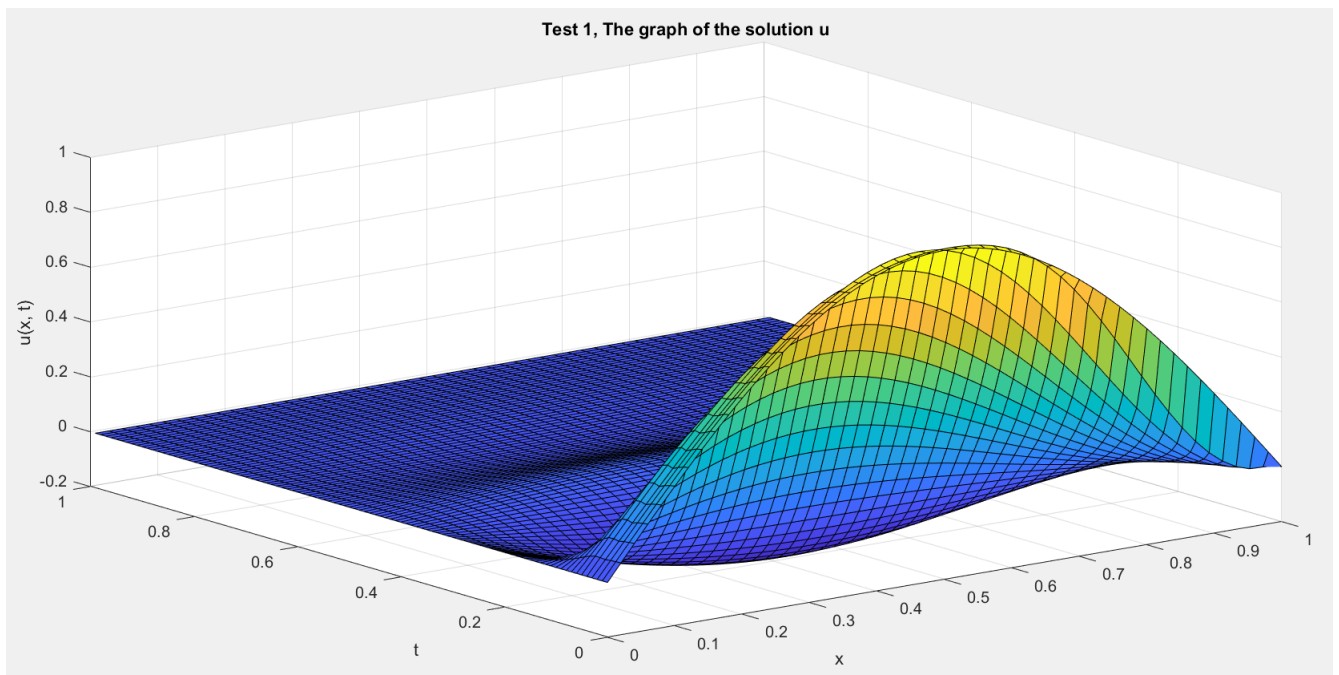

**Figure 5.** Test 1: The solution function $u(x, t)$.

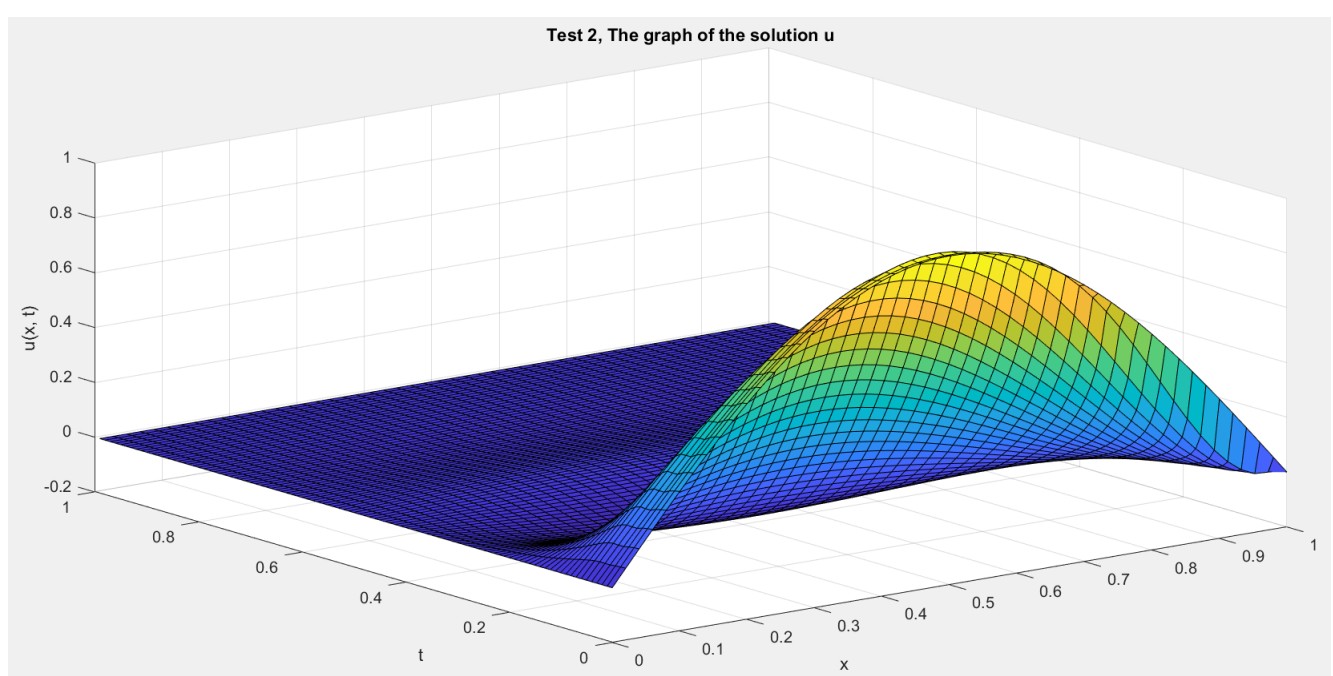

**Figure 6.** Test 2: The solution function $u(x, t)$.

## 6. Conclusions

In this work, we considered a weakly dissipative viscoelastic equation with variable-exponent nonlinearity. We showed that the decay rate of the energy is weaker than that of the relaxation function. An open question is: can we obtain a similar or even weaker decay rate in the absences of the damping term $\Delta u_t$?

**Author Contributions:** Conceptualization, M.M.A.-G. and A.M.A.-M.; methodology, M.M.A.-G. and A.M.A.-M.; software, M.N.; validation, M.M.A.-G., A.M.A.-M. and J.D.A.; formal analysis, M.M.A.-G. and A.M.A.-M.; investigation, M.M.A.-G. and A.M.A.-M.; data curation, M.N.; writing—original draft preparation, J.D.A.; writing—review and editing, M.M.A.-G. and A.M.A.-M.; visualization, A.M.A.-M.; supervision, M.M.A.-G.; project administration, A.M.A.-M.; funding acquisition, A.M.A.-M. All authors have read and agreed to the published version of the manuscript

**Funding:** This research was funded by KFUPM grant number SB20101.

**Acknowledgments:** The authors would like to express their profound gratitude to King Fahd University of Petroleum and Minerals (KFUPM) for its continuous support. The authors also thank the referee for his/her very careful reading and valuable comments. This work was funded by KFUPM under Project #SB201012.

**Conflicts of Interest:** The authors declare that there is no conflict of interest.

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
