# Peer review of "Stability Results for a Weakly Dissipative Viscoelastic Equation with Variable-Exponent Nonlinearity: Theory and Numerics"

_mca, doi:10.3390/mca28010005_

Round 1

Reviewer 1 Report

Review manuscript ID: mca-2108202
Type of manuscript: Article
Title: Stability Results for a Weakly Dissipative Viscoelastic Equation with Variable-exponent Nonlinearity: Theory and Numerics
Authors: Adel M. Al-Mahdi *, Mohammad Al-Gharabli, Maher Noor, Johnson D. Audu
Date: 13 December 2022

The manuscript describes the analytical aspects of an integral-differential equation that has applications in mechanics and physics. The authors also provided illustrations by choosing some examples and calculated the results numerically. Here are some remarks.

* The mathbb G in the abstract is neither introduced nor defined.

* A similar case with the calligraphic A in (1.1).

* Line 34: Use "such as" instead of "like".

* Line 39: It is not clear if alpha is a function.

* Line 52: missing a colon.

* (1.7): missing a space.

* Line 54: a is a positive constant, is this a the same a as a in line 39? Over there it was a function, but not it is a constant. This is really confusing.

* Line 57: too many parentheses.

* Line 57: system

* Line 61: adopt compact citation style, e.g., [1--3].

* Lines 63-64: reword the sentence starting with However.

* In Section 2, there is H on the left-hand side, but there is H^3 inside the set. What do you mean by this?

* Please check whether the correct writing would be "ess inf" and "ess sup".

* Lemma 2.1: p2 is not defined there, perhaps the expression for p star should contain p2 (??)

* Line 87: Confirm that C1(mathbb R^+) is correct.

* In (2.5): confirm that circle means composition. Else, define and explain.

* Line 94: loc should not be italics.

* Line 129: remove a comma.

* (3.17): what does it mean the symbol tilde?

* Line 139: incomplete sentence.

* The curly brackets should be a larger size.

* is >= 0 should be replaced either with mathematical expression or with wording.

* (4.1): what is the function xi(s)? It was mentioned in neither assumption.

* What is the difference between functionals eta and theta? What is the purpose of taking such, and another, different, functional(s)?

* Lines 185-186: comma problem. Please fix that.

* All 2D figures are unclear. Enlarge the labeling, and use a better resolution, particularly for those curves.

Author Response

kindly see the attached files

Reviewer 2 Report

See the enclosed file.

Author Response

kindly, see the attached files
